# Selective NO_2_ Detection of CaCu_3_Ti_4_O_12_ Ceramic Prepared by the Sol-Gel Technique and DRIFT Measurements to Elucidate the Gas Sensing Mechanism

**DOI:** 10.3390/ma16093390

**Published:** 2023-04-26

**Authors:** Rodrigo Espinoza-González, Josefa Caamaño, Ximena Castillo, Marcelo O. Orlandi, Anderson A. Felix, Marcos Flores, Adriana Blanco, Carmen Castro-Castillo, Francisco Gracia

**Affiliations:** 1LabMAM, Department of Chemical Engineering, Biotechnology and Materials, FCFM, Universidad de Chile, Santiago 8370456, Chile; josefamargot4297@gmail.com (J.C.); xcastillo@ing.uchile.cl (X.C.);; 2Department of Engineering, Physics and Mathematics, Sao Paulo State University (UNESP), Araraquara 14801-385, Brazil; marcelo.orlandi@unesp.br (M.O.O.); aafelixy@yahoo.com.br (A.A.F.); 3Physics Department, FCFM, Universidad de Chile, Santiago 8370448, Chile; mflorescarra@ing.uchile.cl

**Keywords:** CCTO, DRIFT spectroscopy, gas sensor, NO_2_ selectivity

## Abstract

NO_2_ is one of the main greenhouse gases, which is mainly generated by the combustion of fossil fuels. In addition to its contribution to global warming, this gas is also directly dangerous to humans. The present work reports the structural and gas sensing properties of the CaCu_3_Ti_4_O_12_ compound prepared by the sol-gel technique. Rietveld refinement confirmed the formation of the pseudo-cubic CaCu_3_Ti_4_O_12_ compound, with less than 4 wt% of the secondary phases. The microstructural and elemental composition analysis were carried out using scanning electron microscopy and X-ray energy dispersive spectroscopy, respectively, while the elemental oxidation states of the samples were determined by X-ray photoelectron spectroscopy. The gas sensing response of the samples was performed for different concentrations of NO_2_, H_2_, CO, C_2_H_2_ and C_2_H_4_ at temperatures between 100 and 300 °C. The materials exhibited selectivity for NO_2_, showing a greater sensor signal at 250 °C, which was correlated with the highest concentration of nitrite and nitrate species on the CCTO surface using DRIFT spectroscopy.

## 1. Introduction

The growing concern around the world for air quality in recent decades has promoted the search for an assured supply of clean air that benefits our health and the environment. In particular, the climate change produced by the increase in the global temperature and attributed to greenhouse gas emissions has encouraged the investigation of new materials that detect gases such as CO_2_, NH_4_ and NO_X_, which are the most influential to the greenhouse effect [1]. Nitrogen oxides, typically NO and NO_2_, are mainly generated by the combustion of fossil fuels and various industrial processes and are the major causes of acid rain and photochemical smog, having a significant influence on air, water and soil pollution [1,2,3]. One of the complications of these gases is that it remains in the atmosphere for approximately 100 years, and its heating potential is approximately 280 times greater than CO_2_ [4].

The need to detect these gases has encouraged a global market size that accounted for USD 3.16 billion in 2022 and is projected to surpass around USD 6.2 billion by 2030 [5]. NO_2_ gas sensors are used to sense the concentrations of different flammable and toxic gases, and there is a growing demand for them in the industrial, automotive, and petrochemical sectors. NO_2_ gas sensors in industrial applications are being used for detecting gas leakage and for monitoring the air quality in different industrial sectors [6,7].

The short-term exposure to concentrations of NO_2_ can cause inflammation of the airways and an increased susceptibility to respiratory infections, chronically weakening the respiratory system and possibly slowing down lung function [2,8]. Due to the harmful effects of NO_2_ to the environment and human activities, strict occupational exposure limits (OEL) have been proposed by governmental agencies, such as the European Scientific Committee on Occupational Exposure Limits (SCOEL), which recommends the hourly NO_2_ concentration threshold as 0.1 ppm (200 μg/m^3^) [9,10]. Therefore, there is an essential need for the development of highly sensitive, lower power consuming, and well-performing selective NO_2_ sensors [1,11].

The direct inspection of the exhaust emissions from engines and industrial processes requires devices that detect NO_2_ at middle temperatures. For this, semiconducting metal oxide, SMO, gas sensors have desirable properties, such as high sensitivity and good stability in harsh environments, and can easily be miniaturized to lower cost and power consumption [12,13,14]. Different n-type SMO-based sensors have been studied and proposed as promising candidates for the selective detection of NO_2_, such as WO_3_ [15,16,17], MoO_3_ [18], ZnO [19] and tin oxides [1,20,21].

The SMO sensing mechanism is based on the change in the electrical resistance of the sensing materials upon the exposure to the target gases [22]. When the gas sensor baseline is air, n-type SMO adsorbs oxygen on the surface, which traps a certain number of free electrons from the conduction band of the oxide. Ambient reducing gases react with the formed oxygen ions, which returns the trapped electrons to the conduction band and results in a decrease in the sensor resistance. On the contrary, oxidizing gases, such as NO_2_, also capture a certain number of free electrons from the conduction band and become negatively charged adsorbates, which results in an increase in the sensor resistance [23,24].

On the other hand, the pseudo perovskite ceramic CaCu_3_Ti_4_O_12_ (CCTO) has attracted much attention due to its extremely high dielectric constant [25], thermal stability and non-ohmic properties [26,27]. In this context and motivated by its remarkable properties, studies about the gas sensor response of CCTO have shown promising properties for the detection of H_2_ [28], O_2_ [24,29,30], NO_2_ [24], H_2_S [31] and humidity [32,33].

This study presents the characterization and gas sensing response to NO_2_, H_2_, CO, C_2_H_2_ and C_2_H_4_ compounds of pure CCTO prepared by the sol-gel technique. The study also concerns the surface reactions responsible for the change in resistance observed during the exposure to oxidizing gases by diffuse reflectance infrared Fourier transform (DRIFT) spectroscopy at different temperatures to identify the sensing mechanism.

## 2. Materials and Methods

Pure CCTO nanoparticles were synthesized by the sol-gel technique. In this synthesis, titanium (IV) isopropoxide Ti(OC_3_H_7_)_4_ (97% Sigma-Aldrich, St. Louis, MO, USA), calcium (II) nitrate tetrahydrate Ca(NO_3_)_2_·4H_2_O (99% Sigma-Aldrich, St. Louis, MO, USA), copper (II) nitrate trihydrate Cu(NO_3_)_2_·3H2O (99% Sigma-Aldrich, St. Louis, MO, USA), ethanol C_2_H_5_OH (JT Baker, Allentown, PA, USA), ethylene glycol C_2_H_6_O_2_ (JT Baker, Allentown, PA, USA) and, citric acid monohydrate C_6_H_8_O_7_·H_2_O (≥99% Sigma-Aldrich, St. Louis, MO, USA) were used as starting materials. All the chemicals were of analytical grade and no further purification was performed. The synthesis steps were the following: (a) An amount of 1.859 g of Ca(NO_3_)_2_·4H_2_O, 5.704 g of Cu(NO_3_)_2_·3H_2_O and 9.923 g of citric acid were dissolved by stirring in 10 mL, 20 mL and 20 mL of ethanol, respectively. Once dissolved, the Cu and citric acid solutions were added to Ca solution and mixed all together under vigorous stirring for 1 h at room temperature. (b) In parallel, a solution of 9.599 mL of Ti(OC_3_H_7_)_4_ was prepared with 30 mL of ethanol, which was stirred for 30 min at room temperature; (c) Then, both solutions were mixed together, and 1.98 mL of ethylene glycol was added to the mixture, which was stirred for 1 h to produce the gelification of the solution. (d) Then, the gel was dried at 90 °C for 12 h, followed by a second drying at 120 °C for 6 h. The resultant porous material was grinded in an agate mortar and, finally, calcined at 800 °C for 3 h in air to obtain the CCTO powders.

The obtained powders after calcination were characterized by X-ray diffraction (XRD) in a D8 Bruker diffractometer using CuKα radiation (λ = 1.5418 Å). The data were collected at room temperature with a step size and scan rate of 0.01° and 0.1 s. The X-ray tube was operated at 40 kV and 30 mA. Rietveld refinement of XRD patterns was performed using TOPAS software for which a pseudo-Voigt function was chosen as a profile function.

The morphology and microstructure of the sample powders were studied using a field emission scanning electron microscope (FEG-SEM; JEOL, Model 7500F, Tokyo, Japan) equipped with an X-ray energy dispersive spectroscopy detector (EDS). The samples were prepared by dispersion in isopropanol and deposited on a Si conductive substrate. The oxidation states of the samples were determined by X-ray photoelectron spectroscopy (XPS) using a Physical Electronics 1257 system with non-monochromatic MgKα radiation operating at 15 kV and 400 W. The spectrum calibration was performed using a binding energy of 284.5 eV, corresponding to the C1s orbital. Spectra were fitted by Multipack software using Gaussian–Lorentzian functions after Shirley-type background subtraction.

Gas sensor test devices were prepared using interdigitated platinum electrodes sputtered with 300 μm thickness spaced by 300 μm over insulating alumina substrates. Metallic tracks on the backside of the substrate were used as the heater element. Gas sensing tests were performed at 100, 150, 200, 250 and 300 °C, and the resistance was monitored using a stabilized high voltage source measuring unit (Keysight 34972A, Technologies, Inc., Santa Rosa, CA, USA) at a constant voltage of 100 mV with 8 s delay per point. The gas sensing response of the CCTO samples was analyzed during cyclic exposure (20 min gas exposure with 60 min recovery) to different concentrations of NO_2_, H_2_, CO, C_2_H_2_ and C_2_H_4_ gases. The baseline was established by dry synthetic air during 12 h, and then exposed to analyte gases in a concentration range between 2 and 100 ppm. To achieve this, certified pre-mixed gas mixtures containing a trace of the test gases diluted in dry air (White Martins, Sao Paulo, SP, Brazil) were mixed with clean dry air using mass flow controllers (MKS, Andover, MA, USA). The total gas flow rate (test gas plus baseline gas) was kept constant (100 sccm) during all tests. More details about this self-heating gas sensing system can be found in Felix et al. [34].

Surface gas reactivity of the CCTO compound was studied by diffuse reflectance infrared Fourier transform spectroscopy, DRIFTS, using an FT-IR spectrometer (Thermo Scientific; FTIR Nicolet iS50 spectrometer, Waltham, MA, USA) containing a narrow-band mercury cadmium telluride (MCT) detector cooled by liquid nitrogen. The CCTO powders were mounted in a diffusion cell chamber from PIKE technologies with a KBr window and temperature control. The powders were heated at 300 °C for 1 h in the presence of dry air with a flow of 20 mL/min to clean the powder surface in similar conditions to the gas sensing experiments. Later, the same flow of a NO_2_/air mixture with a concentration of 0.5% was introduced into the diffusion cell chamber to obtain a significant signal in the DRIFT cell, and FTIR measurements at 300, 275, 250 and 225 °C were performed for 25 min, while spectra were recorded every 5 min with scans between 1000 and 4000 cm^−1^ and with a step of 4 cm^−1^. The signals of NO_2_, NO, He and O_2_ were simultaneously followed by a mass spectrometer (OmniStar™, Pfeiffer; gas analysis system, GmbH, Asslar, Germany). The same study was performed using a gas flow of a NO_2_/He mixture to compare the influence of oxygen on DRIFTS results.

## 3. Results

### 3.1. XRD Characterization

Figure 1a shows the XRD pattern of the CCTO sample. The analysis confirmed the formation of the pseudo-cubic CCTO compound (JCPDS File No. 75-2188), and the residual formation of CuO was also detected. Rietveld refinement (Figure 1b), with a GoF parameter of 1.5, confirmed the mentioned CCTO phase that exhibited a unit cell parameter (*a*) of 0.7397 nm and a crystallite size (*t*) of 72 nm. The refinement also showed that the secondary CuO phase is 3.6 wt.% of the sample.

### 3.2. SEM Results

The microstructure observed by SEM showed the formation of polycrystalline particles with sizes between 1 and 5 μm, as shown in Figure 2. The crystalline grains of the micrometric particles exhibited sizes of about 300 nm. The EDS analysis performed on the CCTO sample indicated the presence of the expected elements, as shown in the inset spectrum in Figure 2.

### 3.3. XPS Results

The XPS analysis of CCTO powders revealed the expected results. To elucidate the surface chemistry and oxidation states of the different elements, high-resolution XPS spectra (HRXPS) were collected for Ca, Cu, Ti and O, as shown in Figure 3.

The analysis of the Ca2p signal (Figure 3a) showed that the HRXPS can be fitted by two Gaussian curves with main binding energies at 346.4 and 347.2 eV, which have been associated with Ca^2+^ [35] and the same oxidation state in superficial CaTiO_3_-like structures [36], respectively. Similar analyses were performed for the other elements and the results are presented in Table 1. The energies found in the Cu spectrum (Figure 3b) evidence two peaks, in addition to the expected satellites for Cu, which were attributed to different Cu-O coordination sites [37]. The lower energy was assigned to Cu^+^, while the higher energy was assigned to species involving Cu^2+^. The Ti2p spectrum (Figure 3c) can also be fitted by two Gaussian curves, indicating the presence of different valence states. The binding energies can be assigned to Ti^4+^ (458.6 eV) and Ti^3+^ (457.9 eV) [37,38]. The O1s spectra (Figure 3d) is decomposed in two curves, for which peaks are assigned as lattice oxygen (O^2−^) [35] and chemisorbed oxygen species on the surface, like in the case of $O_{2}^{-}$ [31].

It was also measured the area under the curve of the fitted curves was also measured to estimate the relative amount between the various atomic species, for which results are also presented in Table 1. From this analysis, it follows that most of the calcium is in the Ca^2+^ state, whereas most of the copper is in the Cu^2+^ state with a slight portion of Cu^+^. The 23% of Ti^3+^ ions calculated from the Ti signal are associated with the oxygen vacancies in CCTO ceramics [37,38].

### 3.4. Gas Sensing Response

The gas sensing response was tested for different concentrations of NO_2_, H_2_, CO, C_2_H_2_ and C_2_H_4_. The material exhibited the most significant gas sensor response for NO_2_, while for the other gases the response was very low. Figure 4 shows the sensor response of the CCTO sample at 250 °C for different NO_2_ concentrations, and a reversible sensor response down to the lowest levels of gas exposure can be seen. The sensor signal (SS) was defined as the ratio of the sensor resistance measured when exposed to the target gas (R_gas_) to the resistance in the baseline gas (R_air_), i.e., R_gas_/R_air_. It can be observed that the SS grows as the concentration of NO_2_ increases from 2 to 100 ppm. These results are similar to those published by Felix et al. [24] that reported an SS close to 7 when material was exposed to 100 ppm of NO_2_ at 300 °C in CCTO thin films prepared by the polymeric precursor method.

Additional measurements at different temperatures showed that the sensor signal is higher at 250 °C for almost all the tested concentrations, which means that the SS was temperature dependent, as shown in Figure 5. The increasing resistance of CCTO samples under the oxidant atmosphere of NO_2_ is consistent with the n-type behavior of the SMO sample, which was also reported by Parra et al. [29] and by Felix et al. [24] for a CCTO ceramic prepared by the sol-gel technique. This electronic response can be attributed to the oxygen vacancies deduced from the Ti^3+^/Ti^4+^ ratio measured by XPS. Oxygen vacancies carry an effective charge of +2e, which is neutralized by 3d electrons on the titanium atoms, forming two Ti^3+^ ions for every oxygen vacancy. At low temperatures, the oxygen vacancies and the Ti^3+^ ions are bound by a small energy of 0.1–0.2 eV, which is sufficiently large, so that only a few of the defects are separated. Electrons associated with the unattached Ti^3+^ ions are responsible for conduction, making use of the narrow 3d conduction band [39].

The range of selectivity of a gas sensing material is an important parameter used to evaluate practical applications. Figure 6a illustrates the selectivity of CCTO to NO_2_, H_2_, CO, C_2_H_2_ and C_2_H_4_ (10 ppm) at 250 °C, represented by the ratio between the SS in the presence of NO_2_ and the SS under the interferent gas, i.e., ${SS}_{{NO}_{2}}/{SS}_{interferent}$. Under these conditions, the response of the CCTO sample was more than 2.5-fold larger for NO_2_ than for the other gases. The radar chart of Figure 6b depicts the SS at 250 °C for the various gas concentrations tested, and it confirms that the selectivity of CCTO increases with the gas concentration.

### 3.5. DRIFTS Measurements

The most accepted model for gas detection in SMO materials is related to the transfer of free charge carriers between the absorbed molecules and the semiconductor surface [23]. This model proposes that, in air atmosphere, n-type semiconductors have a depletion layer at the surface produced by the electron transfer from the surface to the chemosorbed oxygen molecules. In other words, the absorbed oxygen molecules trap electrons from the oxide surface by ionization and become $O_{2}^{-},\;O^{-}$ or $O^{2-}$ depending on the temperature [40]. The density of electrons in the depletion layer decreases with the concentration of chemisorbed oxidizing analyte gases at the surface, leading to an increase in surface resistance and, consequently, sensor resistance [23,24]. Thus, the variations of the increase in the electrical resistance at different temperatures should be related to the concentration of oxygen adsorbates and the nitride species on the ceramic surface.

IR DRIFT spectroscopy is an excellent tool to keep track of changes at the surface induced by the interaction between a target gas and the sensing material. Figure 7a shows the DRIFTS curves measured at 250 °C under NO_2_/air and NO_2_/He mixtures, considering that the wavenumber ranges that are of interest are below 1900 cm^−1^. The spectrum obtained in the exposure of NO_2_/air exhibits the typical range between 1600 and 1200 cm^−1^ that has been attributed to various nitrite, ${NO}_{2}^{-}$, and nitrate, ${NO}_{3}^{-}$, species, which could be from molecularly adsorbed NO_2_ [41]. The peak at 1630 cm^−1^ is associated with the vibrational band of gaseous NO_2_ [42], while the peaks at 1600 and 1569 cm^−1^ seem to arise from asymmetric and symmetric stretching vibrations of nitro species, respectively [22]. Overall, the bands below 1580 cm^−1^ correspond to the N–O stretching vibrations of surface nitrate species [43]. In contrast, despite that, in the spectrum of the experiment with NO_2_/He, peaks at 1630 and 1600 cm^−1^ can be identified, the intensity of the bands below 1580 cm^−1^ are much lower. This will be related to lower concentrations of nitro species on the surface, as will be shown below.

Figure 7b depicts the DRIFT spectra at different temperatures obtained under the NO_2_/air mixture. It is observed that the spectra are similar, although with different intensities for the nitrite and nitrate bands, with the highest signals measured at 250 °C. The bands observed between 1580 and 1450 cm^−1^ can be attributed to the adsorbed species on the CCTO surface, which is promoted by the oxygen present in the air mixture, given the higher amounts in those experiments. According to Beer’s law, the integral absorbance of the IR spectra is related to the surface concentration of adsorbed species on the surface [44]. Thus, using the software OMNIC™ Specta (Thermo-Fisher), the integral absorbance as the area under the curve was calculated to be between 1650 to 1400 cm^−1^ of the DRIFTS spectrum measured in the NO_2_/air and NO_2_/He mixture. The subtraction between the former and the latter is plotted in the inset image of Figure 7b as the differential integral absorbance among NO_2_/air and NO_2_/He atmospheres. The curve reaches a maximum value at 250 °C, which decreases at higher temperatures. This confirms that at 250 °C the maximum concentration of NO_2_ molecules is adsorbed as nitrite and nitrate species on the CCTO surface. This is well correlated with the highest sensing response plotted in Figure 5, since a large concentration of such adsorbates is essential to produce the change in electrical resistance.

The exact identification of peaks in the nitrate region is not straightforward as the different nitrate species (i.e., monodentate nitrates, bridging monodentate, chelating and bridging bidentate nitrates) have overlapping vibrations [43]. Ueda et al. [22] studied the NO_2_-sensing properties of In_2_O_3_, which is also an n-type semiconductor gas sensor. They showed that the nitrite ${NO}_{2}^{-}$ species form monodentate, chelating bidentate and nitro compounds anchored to the In atoms. Similarly, DRIFT studies of the catalytic reduction of NO_X_ on a zeolite-supported Cu catalyst have shown that the group of peaks in the 1650–1500 cm^−1^ wavenumber region after the exposure to NO_X_ and O_2_ can be assigned to different configurations of surface nitrates bonded to Cu atoms [45,46].

The origin of the n-type behavior in CCTO has been explained due to structural defects, particularly oxygen vacancies, that produce a moderate conductivity at elevated temperatures [31]. In n-type SMO, the electron concentration is mainly determined by the concentration of stoichiometric defects, such as oxygen vacancy, and it is believed that the gas sensitivity of n-type SMO is proportional to the concentration of oxygen vacancy related defects [47]. This generates electrons in the conduction band, which, in the surface, interact with oxygen to form chemisorbed oxygen ions. Similarly, Roso et al. [48] proposed that the sensing mechanism of NO_2_ in In_2_O_3_ at 350 °C takes place first for the formation of the nitrite specie:(1)$${NO}_{2,\mathrm{gas}}+e^{-}⇌{NO}_{2,\mathrm{ads}}^{-}$$

Thus, it is proposed that in CCTO, according to the evidence, the reaction of Equation (1) would occur among NO_2_ molecules and electrons in a 3d conduction band promoted by oxygen vacancies [39]. This reaction is responsible for the increasing resistance of the sensor proportional to the NO_2_ concentration. The proposed sensing mechanism reaches its maximum at 250 °C due to the increased conductivity of the ceramic at that temperature [31]. The adsorbed nitrate species would be bonded to the Cu atoms [45,46] of the CCTO structure, as suggested by DRIFT spectra.

Since the highest concentration of NO_2_ molecules adsorbed as nitrite is at 250 °C on the CCTO surface, the decreasing response at 300 °C is explained by the lower adsorption of the nitro molecules on the surface, which reduces the change in the electrical resistance.

## 4. Conclusions

The sensing response of the CaCu_3_Ti_4_O_12_ ceramic prepared by the sol-gel technique was studied for different gases. The gas sensing response of the samples exhibited a high selectivity of NO_2_ compared to H_2_, CO, C_2_H_2_ and C_2_H_4_ analytes, showing the higher sensor signal at 250 °C. DRIFTS measurements confirm that at 250 °C the maximum concentration of NO_2_ molecules is adsorbed on the CCTO surface, which would react with electrons in the conduction band of nitrite species. This is well correlated with the highest sensing response obtained at that temperature since a large concentration of such adsorbates is essential to produce the change in electrical resistance. DRIFT measurements contribute to monitoring the sensing mechanisms under operating conditions, which allows deeper insights into the chemical sensing phenomenology and, consequently, to the development of superior chemical sensor devices. Additional characterizations, such as long-time stability and humidity effects, are the next steps of our research to explore the potential commercial applications on this CCTO sensor.

## Figures and Tables

**Figure 1 materials-16-03390-f001:**
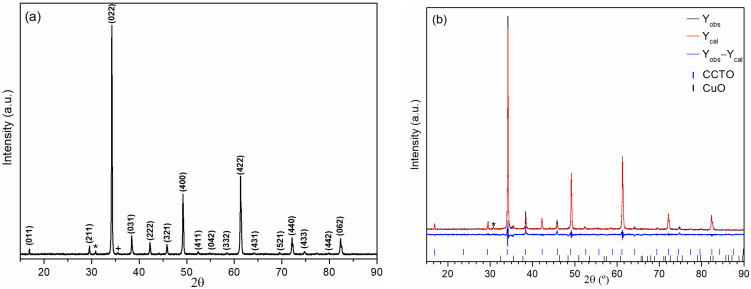
(**a**) XRD pattern of CCTO sample, (*) *K_α_* peak from (022) reflection. (+) Peak from CuO phase; (**b**) Rietveld refinement of XRD CCTO pattern.

**Figure 2 materials-16-03390-f002:**
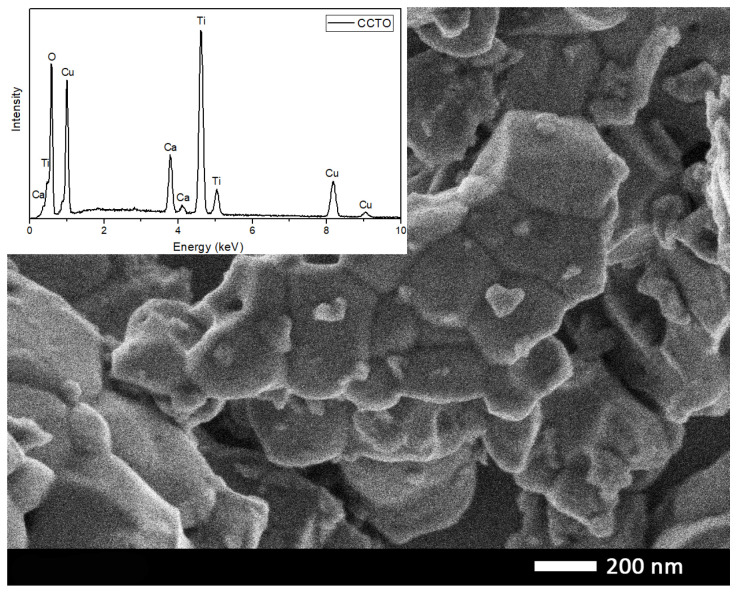
SEM image of CCTO sample obtained by secondary electrons detector. Inset: EDS analysis of CCTO particles.

**Figure 3 materials-16-03390-f003:**
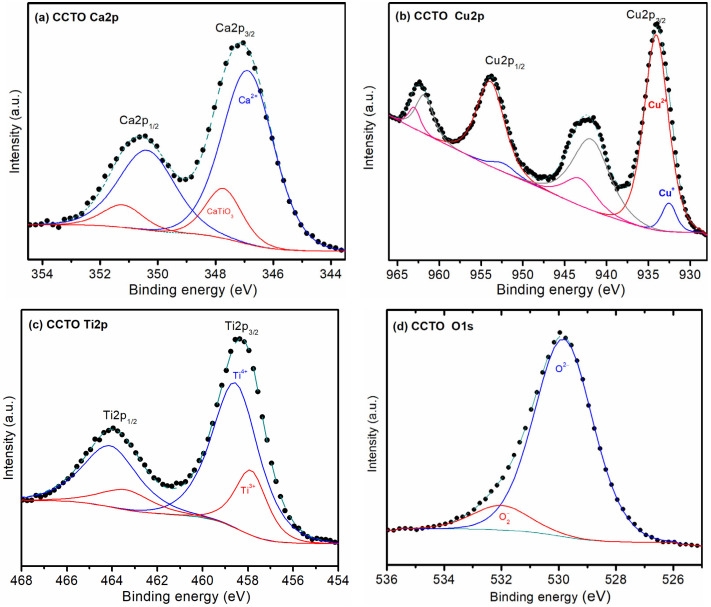
High-resolution XPS spectra of the different elements present in sample CCTO: (**a**) Ca2p, (**b**) Cu2p, (**c**) Ti2p, and (**d**) O1s.

**Figure 4 materials-16-03390-f004:**
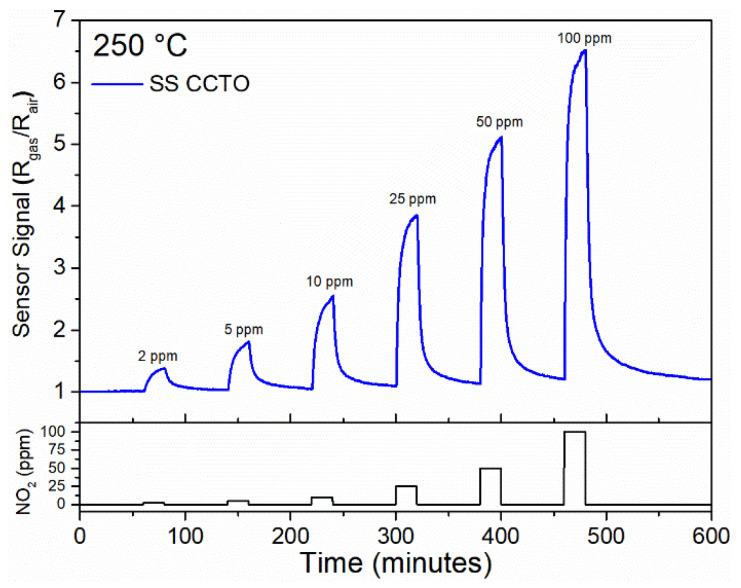
Gas sensing response of CCTO sample as a function of the NO_2_ concentration at 250 °C.

**Figure 5 materials-16-03390-f005:**
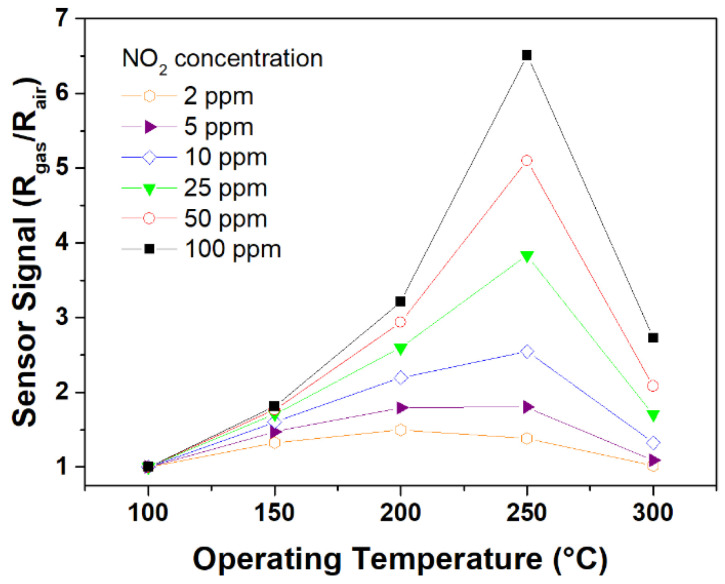
Gas sensing response plots as a function of the temperature of different NO_2_ concentrations.

**Figure 6 materials-16-03390-f006:**
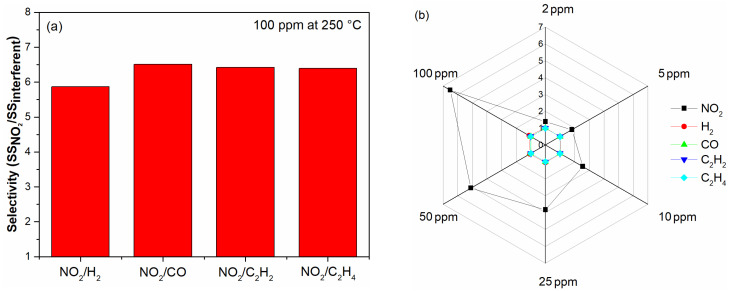
(**a**) Gas selectivity at 10 ppm and 250 °C of CCTO sample; (**b**) sensor signal of CCTO sample at 250 °C for the concentrations of different gases.

**Figure 7 materials-16-03390-f007:**
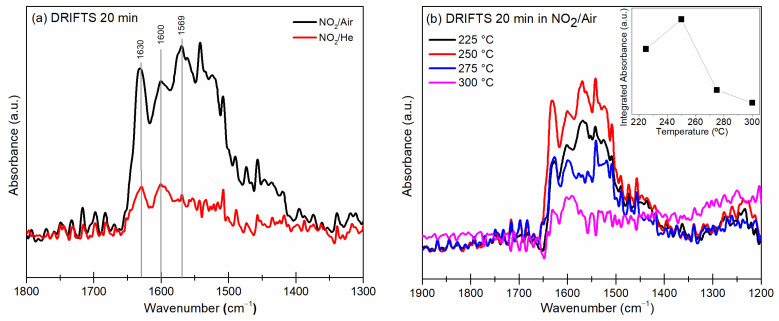
DRIFT spectra of CCTO powders (**a**) at 250 °C in NO_2_/air and NO_2_/He mixture atmospheres and (**b**) in NO_2_/air mixture at different temperatures and differential integrated absorbance (inset).

**Table 1 materials-16-03390-t001:** Binding energies, area under the curve, and oxidation state as obtained from HRXPS spectra.

Sample	CCTO		
Element	Binding Energy ^1^	Area ^2^	Oxidation State ^3^
Ca	346.4	79.1%	Ca^2+^
	347.2	20.9%	CaTiO_3_
Cu	932.5	8.0%	Cu^+^
	934.1	92.0%	Cu^2+^
Ti	457.9	23.3%	Ti^3+^
	458.6	76.7%	Ti^4+^
O	529.8	89.2%	O^2−^
	532.0	10.8%	$O_{2}^{-}$

^1^ Binding energy obtained from the fitted curve in the HRXPS spectra. ^2^ Relative percentage of the area under the curve normalized to the fitted curve. ^3^ Representative oxidation state or species assigned to the binding energy.

## Data Availability

No new data were created or analyzed in this study. Data sharing is not applicable to this article.

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
