# Peer review of "Selective NO2 Detection of CaCu3Ti4O12 Ceramic Prepared by the Sol-Gel Technique and DRIFT Measurements to Elucidate the Gas Sensing Mechanism"

_materials, 2023, doi:10.3390/ma16093390_

Round 1
Reviewer 1 Report
I have reviewed the manuscript entitled "Selective NO2 detection of CaCu3Ti4O12 ceramic prepared by the sol-gel technique and DRIFT measurements to elucidate the gas sensing mechanism". The manuscript reports the gas sensing properties of the CaCu3Ti4O12 compound prepared by the sol-gel technique. The authors have characterized the microstructure and elemental composition of the material using scanning electron microscopy (SEM) and X-ray energy dispersive spectroscopy (EDS), respectively. Additionally, the elemental oxidation states of the samples were determined by X-ray photoelectron spectroscopy (XPS). The gas sensing response of the samples was performed for different concentrations of NO2, H2, CO, C2H2 and C2H4 at temperatures between 100 and 300 °C. The materials exhibited selectivity for NO2, showing the greatest sensor signal at 250 °C, which was correlated by DRIFT spectroscopy with the highest concentration of nitrite and nitrate species on the CCTO surface.
The manuscript is well-written and well-presented. However, I have a few concerns that need to be addressed. Firstly, the authors should describe in more detail the sol-gel preparation method, the synthesis conditions, the post-treatment of the samples, the particle sizes and the surface area of the sensor.
Secondly, I would like to see the authors provide more information on the gas sensing mechanism. They should discuss the possible interactions between NO2 and the CCTO surface. They should also discuss the role of the oxygen vacancies in the sensing mechanism. This information would help to further elucidate the gas sensing mechanism of CCTO.
Additionally, the authors should provide more discussion on the long-term stability of the CCTO sensor.
Finally, the authors should consider revising the conclusion to provide more insights into the potential applications of the material and future research directions.
Overall, the manuscript presents interesting results and contributes to the field of gas sensing materials. However, the concerns I have mentioned above need to be addressed to strengthen the manuscript further.
Minor editing of English language required.
Reviewer 2 Report
The authors reported the structural and gas sensing properties of the CaCu3Ti4O12 compound prepared by the sol-gel technique. The gas sensing response of the samples was measured for different concentrations of NO2, H2, CO, C2H2 and C2H4 at temperatures between 100-300 °C. Synthesis steps and structural characterization of pseudo-perovskite ceramic CaCu3Ti4O12 (CCTO) material are reported in detail.
1) Were the I-V measurements of the sensor made in dry air and NO2 gas flow? Have the sensor response values been compared with the I-V results?
2) If the NO2 concentrations in Figure 4 are written over the peaks, they will be more pronounced.
3) The gas sensing response plots in Figure 5 show a decrease after 250 oC. It should be discussed why it decreases after the peak that occurs at 250 oC.
4) I think that some new references may be given in the manuscript such as following or similar:
i. https://www.sciencedirect.com/science/article/abs/pii/S0925400521009990
ii. https://iopscience.iop.org/article/10.1149/2162-8777/ababdd/meta
iii. https://pubs.acs.org/doi/full/10.1021/acsami.1c06546?casa_token=3h1x-up78XcAAAAA%3AP-HHrYL_atLwrDpI8_7GmAH5OusfNRgH3rJZDy6K7Vti4IVg7LylBd_-mZh02YmpAgEA0zY1NeI
iv. https://www.sciencedirect.com/science/article/pii/S0925838822022228?casa_token=h3YRuUAs00QAAAAA:7ofSO3dNakPenpyINySzrFRUZ1dolBVmuGnBodAvJdQ4ZG1J2E7ELkchjZfzKtySbXhqPA
Reviewer 3 Report
This work deals with the structural and gas sensing properties of the CaCu3Ti4O12 compound prepared by the sol-gel technique. The microstructural and elemental composition analysis were carried out using scanning electron microscopy and X-ray energy dispersive spectroscopy, respectively, while the elemental oxidation states of the samples were determined by X-ray photoelectron spectroscopy. Moreover, DRIFT spectroscopy was applied.
THE WHOLE WORK IS INTERESTING.
POINTS FOR IMPROVEMENT:
1. The intrduction focus too much on the pollution effect of NO2. This part could be shorten without removing references. In addition a few paragraphs could be written for the a) general status of gas sensor area b) for commercial sensors for NO2. An additional literature review is required.
2. Is the proposed sensor a commercial available one? What are the limitations for extensive applications? If it is commercially available what is the originality of this work?
3. Please explain in brief how the results of this work could be applied to optimize sensors.
Round 2
Reviewer 1 Report
I have reviewed the revised manuscript entitled “Selective NO2 detection of CaCu3Ti4O12 ceramic prepared by the sol-gel technique and DRIFT measurements to elucidate the gas sensing mechanism” and I am now satisfied with the authors’ responses to my previous concerns.
The authors have taken the necessary steps to address the issues raised in my previous review and have made the appropriate revisions to their manuscript. In my opinion, the revised manuscript is now suitable for publication in your esteemed journal. Therefore, I would like to recommend its acceptance for publication in Materials.